

# Predictive factors associated with acute radiation dermatitis in patients with breast cancer: a retrospective cohort study

Rattanaporn Nanthong[1,2], Sunanta Tungfung[2], Kamonwan Soonklang[1,3] and Wiriya Mahikul[1]

[1] Princess Srisavangavadhana College of Medicine, Chulabhorn Royal Academy, Bangkok, Thailand
[2] Department of Radiation Oncology, Chulabhorn Hospital, Bangkok, Thailand
[3] Chulabhorn Learning and Research Centre, Chulabhorn Royal Academy, Bangkok, Thailand

Corresponding author
Wiriya Mahikul,
wiriya.mah@cra.ac.th

## ABSTRACT

**Background**. Radiation therapy (RT) is a primary postsurgical treatment for breast cancer; however, it can cause acute radiation dermatitis (ARD), which can severely impair quality of life. The aim of this study was to identify predictive factors associated with moderate to severe ARD.

**Materials and Methods**. In this retrospective analysis, we utilized data from Chulabhorn Hospital's Health Information System that was collected between January 2017 and December 2022. A radiation oncology specialist assessed ARD in a cohort of 635 patients using the Radiation Therapy Oncology Group (RTOG) ARD grading scale. The patients were classified into two groups based on the maximum grade recorded: mild (grade < 2) and moderate to severe (grade ≥ 2). Various factors were examined, including demographic characteristics (age, body mass index (BMI), comorbidities) and treatment-related variables (surgical history, adjuvant chemotherapy, hormone therapy, targeted therapy, fractionation, boost treatments, and bolus application). Logistic regression was used to perform the statistical analysis.

**Results**. Among the 635 patients, the average age was 54.2 ± 10.9 years, and 32% were classified as having moderate to severe ARD. Multiple logistic regression analysis identified BMI ≥ 30 kg/m$^2$ (adjusted odds ratio (AOR) = 2.33; 95% confidence interval (CI) [1.36–3.98]; $p$-value = 0.002), localized boost treatments (AOR = 2.09; 95% CI [1.08–4.06]; $p$-value = 0.029), and bolus application (AOR = 2.08; 95% CI [1.02–4.24]; $p$-value = 0.044) as significant risk factors for moderate to severe ARD. Conversely, hypofractionated RT (AOR = 0.31; 95% CI [0.16–0.57]; $p < 0.001$) and hormonal therapy (AOR = 0.60; 95% CI [0.42–0.86]; $p$-value = 0.005) were associated with a decreased risk. However, radiation to both the primary site and regional lymph nodes (AOR = 0.81; 95% CI [0.41–1.59]; $p$-value = 0.538) and targeted therapy (AOR = 0.72; 95% CI [0.43–1.20]; $p$-value = 0.210) did not significantly affect the risk of moderate to severe ARD.

**Conclusions**. We have identified key risk factors for moderate to severe ARD, including obesity and treatment modalities such as localized boost treatments and bolus application. Hormone therapy and hypofractionated RT appear to reduce ARD severity. These findings have implications for the development of treatment plans and the mitigation of the risk of ARD in patients undergoing RT.

## INTRODUCTION

Breast cancer poses a significant global public health challenge (*Wilkinson & Gathani, 2022*). It is the leading cause of both cancer incidence and mortality among women, with approximately 2.26 million new cases and nearly 685,000 deaths annually (*Wilkinson & Gathani, 2022*). In Thailand, breast cancer ranks first in both cancer incidence and mortality rates among women, mirroring global statistics and underscoring its critical impact on both national and international levels (*Suwankhong et al., 2023*; *Virani et al., 2018*). At Chulabhorn Hospital, around 200–300 breast cancer patients receive radiation therapy (RT) annually, contributing to thousands of RT sessions each year. The hospital consistently treats breast cancer as the most common cancer in its radiation oncology department.

Breast cancer originates from cellular abnormalities in the terminal duct lobular unit (TDLU) or glands that lead to uncontrolled cell division (*Metzger-Filho et al., 2019*). Most breast cancers are adenocarcinomas, of which the breast ducts account for 85% of instances and the lobular epithelium for 15% (*Feng et al., 2018*). A common occurrence of the natural progression of the disease is the spread of carcinomatous cells *via* the lymphatic system to nearby lymph nodes and distant organs, such as the bones, lungs, liver, and brain (*Rahman & Mohammed, 2015*). Breast cancer is primarily diagnosed in women; men account for less than 1% of breast cancer cases (*Konduri et al., 2020*). There is a range of medical treatments available for breast cancer, including surgery, adjuvant chemotherapy, radiation therapy (RT), hormonal therapy, and targeted therapy (*Mee et al., 2023*). Each modality plays an essential role in controlling disease progression and improving patient outcomes.

Despite the advances made in the development of therapeutic options for breast cancer, treatment-related side effects, particularly acute radiation dermatitis (ARD), remain prevalent and can significantly impact patients' quality of life (*Xie et al., 2021*). Acute radiation dermatitis is characterized by edema, endothelial cell changes, and various epidermal and dermal alterations, including inflammation, apoptosis, and necrosis mediated by lymphocytes and cytokines. Cell death typically occurs after one to five cycles of radiation division. Radiation-induced damage to chromosomal DNA leads to cell destruction through apoptosis, necrosis, and mitotic failure (*Lee et al., 2009*). Up to 95% of postsurgical breast cancer patients experience ARD, which can lead to physical discomfort and emotional distress (*Iacovelli et al., 2020*; *Sherman & Walsh, 2022*). The severity of ARD has been found to depend on radiation-related factors, such as the amount of radiation received (total dose of radiation), the number of radiation sessions, the duration of the treatment, and the energy, radiation techniques, and bolus doses applied (*Chan et al., 2014*). The volume of irradiated tissue and the radiation sensitivity of the involved tissue are also contributing factors (*Xie et al., 2021*). Other patient-related factors, such as age, smoking, comorbidities, and concurrent chemotherapy treatments, may also influence possible dermatologic reactions to RT (*Wei et al., 2019*). The consequences of RT and

the resultant side effects include discomfort , difficulty sleeping, anxiety, depression, and body image issues, which can lead to discomfort, anxiety about clothing, and disruption of daily routines and the patient's social life (*Almeida et al., 2023*). When acute dermatitis progresses to a severe stage, it can cause pain and negatively impact the patient's quality of life (*Bashyam et al., 2021*). Some patients may need to pause their treatment due to side effects, and the most serious consequences of side effects are treatment refusal and failure to follow up, which affect cancer survival rates (*Spałek, 2016*). Hence, researchers and healthcare providers are increasingly focused on understanding and mitigating such side effects to improve patient care.

The severity of ARD varies among patients, even among those with identical conditions and treatment regimens. These factors influencing the severity of ARD have been identified, but there is still controversy regarding the results. Elucidating these factors is crucial to improving breast cancer treatment and optimizing personalized patient care strategies. Thus, the objective of this research was to investigate predictive factors that influence the severity of ARD in breast cancer patients undergoing RT.

## MATERIALS AND METHODS

### Study sample

This retrospective study was conducted with data collected between January 2017 and December 2022 and included 635 breast cancer patients who underwent either mastectomy or breast-conserving surgery, followed by postoperative RT at Chulabhorn Hospital. During the RT sessions, the skin of these patients was closely monitored, evaluated, and documented in the Hospital Information System (HIS) by a radiation oncology specialist. Written informed consent was obtained from all patients. All patients underwent weekly follow-up to monitor and document any changes in their skin condition throughout the course of treatment.

The inclusion criteria used to select patients for this study were as follows: female, aged $\geq$ 18 years, diagnosed with primary breast cancer, underwent curative treatment on both concurrent hormonal therapy and adjuvant chemotherapy, completed the full course of radiation therapy (RT). Patients received either conventional fractionated radiation therapy (50 Gy in 25 fractions) or hypofractionated radiation therapy (42.4 Gy in 16 fractions), with or without a sequential boost , and evaluated for ARD by a specialist in radiation oncology. Patients were excluded from the study if they had bilateral breast cancer, had metastatic breast cancer originating from another primary tumor, had undergone reirradiation at the same site, had an infection in the irradiated area, were receiving concurrent chemoradiation, or had incomplete or missing data.

Patient data were obtained from Chulabhorn Hospital's HIS and included information on personal, demographic, and treatment-related factors. Data were obtained on age, body mass index (BMI), comorbidities (*e.g.*, diabetes and hypertension), type of surgery, adjuvant chemotherapy, hormonal therapy during RT, targeted therapy during RT, RT site, RT fractionation, boost treatments, and bolus application. Additional relevant information was obtained from patient clinical records.

**Table 1  The acute radiation dermatitis grading scale developed by the radiation therapy oncology group (RTOG).**

| Grade | Description |
|---|---|
| Grade 0 | No change from baseline skin condition |
| Grade 1 | Mild erythema (redness) and dry desquamation (peeling) |
| Grade 2 | Moderate to brisk erythema, patchy moist desquamation mostly confined to skin folds and creases, or moderate edema (swelling) |
| Grade 3 | Moist desquamation outside of skin folds, bleeding induced by minor trauma or abrasion |
| Grade 4 | Skin necrosis or ulceration of full-thickness dermis, spontaneous bleeding from the involved site; requires local wound care and may require skin grafting |

## Radiation therapy

Each patient's treatment plan was developed based on established National Comprehensive Cancer Network (NCCN) guidelines (Moo et al., 2018). Initially, the clinician created the RT plan and delineated the treatment target based on the patient's specific condition. Subsequently, the medical physicist planned the dose distribution. The results were then reported to the clinician for verification and implementation. RT generally commenced 4–6 weeks after surgery or the completion of adjuvant chemotherapy. In this study, most patients received three-dimensional conformal radiation therapy (3DCRT), while a subset received intensity-modulated radiation therapy (IMRT). A 6–10 MV photon linear accelerator (Varian Medical Systems, Palo Alto, CA, USA) was used to deliver the RT. Patients underwent adjuvant RT using either conventional (50 Gy/25 fractions) or hypofractionated (42.56 Gy/16 fractions) regimens, with or without sequential boost irradiation and/or bolus. Patients with confirmed lymph node metastasis received irradiation of the regional lymph nodes. Individual tumor irradiation boosts, which provided an additional 10–16 Gy in 5–8 fractions to the tumor bed, were administered by the clinician.

## Radiotoxicity evaluation

Assessing ARD is a critical component of patient management, and it is vital to ensure that the evaluation and classification of ARD are precise. The Acute Radiation Dermatitis Grading Scale (Table 1), developed by the Radiation Therapy Oncology Group (RTOG), is the most commonly used clinician-assessed scoring criteria for this purpose (Huang et al., 2015). The scale is used to categorize the clinical symptoms of dermatitis into five grades, ranging from no skin changes (grade 0) to severe ulcerative tissue necrosis or hemorrhage (grade 4). The skin of the participants was evaluated weekly by a radiation oncology specialist throughout the treatment course using this scale, and the results were recorded in the HIS. In this study, patients were categorized into two groups based on the maximum grade recorded: mild (grade < 2) and moderate to severe (grade $\geq$ 2).

## Ethical considerations

The study was conducted in accordance with the guidelines of the Declaration of Helsinki. Before the commencement of this study, ethical approval was granted by the Human

Research Ethics Committee of Chulabhorn Royal Academy, Thailand (code: EC 055/2567). The Institutional Review Board of all participating institutions approved the study. Data were encrypted prior to the analysis at the statistical office, where each patient was assigned a unique identifier. This identifier eliminates the possibility to trace the patient's identity. The final protocol of this study, including the final version of the subject informed consent form, had been approved by the Human Research Ethics Committee of Chulabhorn Royal Academy. Written informed consent was obtained from all patients.

## Statistical analysis

All study variables were subjected to descriptive statistical analysis. Categorical data are expressed as frequency (percent), and quantitative data are expressed as mean $\pm$ standard deviation (SD). The association of each variable with ARD was analyzed using the $\chi^2$ test. We used statistical software to conduct multiple analyses and a binary logistic regression model to identify potential influencing factors for all patients. Considering relevant factors, variables with a *p*-value below 0.05 in the univariate logistic regression analysis, along with other clinically significant variables, were included in the multivariable logistic regression analysis with 95% confidence intervals (CIs). Univariate logistic regression models, also using 95% CIs, were applied to estimate univariate odds ratios (ORs) and adjusted odds ratios (AORs) in the multivariable analysis as previously described in *Poosiripinyo et al. (2023)*. The data analysis was performed using the STATA/MP program (version 18; Stata Corp LLC, College Station, Texas, USA).

# RESULTS

## Incidence of acute radiation dermatitis

Among the 635 patients included in this study, 431 patients (68%) experienced ARD with a maximum grade < 2, which indicated that they exhibited mild reactions, such as erythema and dry desquamation. In contrast, 204 patients (32%) developed more severe reactions, classified as grade $\geq$ 2. These included tender erythema and wet desquamation.

## Patient demographics and treatment characteristics

The demographic and treatment characteristics of the patients are summarized in Table 2. The average age of the cohort was 54.2 $\pm$ 10.9 years. When the BMI data were analyzed, it was found that 374 patients (58.9%) had a BMI < 25 kg/m$^2$, 189 patients (29.76%) were classified as overweight, and 72 patients (11.34%) were considered obese. In terms of comorbidities, 77 patients (12.13%) had diabetes, while 153 patients (24.09%) had hypertension. The patients' surgical history data revealed that 366 patients (57.64%) underwent mastectomy and that 269 patients (42.36%) had breast-conserving surgery. Regarding pre-RT treatment, most of the patients in the cohort (480 patients; 75.59%) received adjuvant chemotherapy before RT. Additionally, hormone therapy was administered to 328 patients (51.65%) during their radiation treatment, while only 85 patients (13.39%) received targeted therapy. From an RT perspective, 398 patients (62.68%) received radiation targeting both the primary tumor site and the regional lymph nodes, while the remaining 237 patients (37.32%) received RT directed solely at the primary site. In terms of the radiation regimens

**Table 2  Clinical characteristics of the patients included in this study.**

| Variables | Category | Total (N = 635) | Maximum grade < 2 (N = 431) | Maximum grade ≥ 2 (N = 204) | p-value |
|---|---|---|---|---|---|
| Age (years) | Mean (SD) | 54.176 (10.976) | 54.652 (10.888) | 53.172 (11.119) | 0.113 |
| | ≤60 | 448 (70.55%) | 297 (68.91%) | 151 (74.02%) | 0.187 |
| | >60 | 187 (29.45%) | 134 (31.09%) | 53 (25.98%) | |
| BMI (kg/m$^2$) | Mean (SD) | 24.291 (4.183) | 25.222 (4.972) | 24.590 (4.469) | 0.014 |
| | <25 | 374 (58.90%) | 268 (62.18%) | 106 (51.96%) | 0.003 |
| | 25–29.90 | 189 (29.76%) | 126 (29.23%) | 63 (30.88%) | |
| | ≥30 | 72 (11.34%) | 37 (8.58%) | 35 (17.16%) | |
| Diabetes | | | | | |
| | No | 558 (87.87%) | 318 (88.40%) | 177 (86.76%) | 0.556 |
| | Yes | 77 (12.13%) | 50 (11.6%) | 27 (13.24%) | |
| Hypertension | | | | | |
| | No | 482 (72.91%) | 328 (76.10%) | 154 (75.49%) | 0.866 |
| | Yes | 153 (24.09%) | 103 (23.9%) | 50 (24.51%) | |
| Surgery type | | | | | |
| | Mastectomy | 366 (57.64%) | 234 (54.29%) | 132 (64.71%) | 0.013 |
| | Breast-conserving surgery | 269 (42.36%) | 197 (45.71%) | 72 (35.29%) | |
| Adjuvant chemotherapy | | | | | |
| | No | 155 (24.41%) | 113 (26.22%) | 42 (20.59%) | 0.123 |
| | Yes | 480 (75.59%) | 318 (73.78%) | 162 (79.41%) | |
| Hormonal therapy | | | | | |
| | No | 307 (48.35%) | 190 (44.08%) | 117 (57.35%) | 0.002 |
| | Yes | 328 (51.65%) | 241 (55.92%) | 87 (42.65%) | |
| Targeted therapy | | | | | |
| | No | 550 (86.61%) | 373 (86.54%) | 177 (86.76%) | 0.939 |
| | Yes | 85 (13.39%) | 58 (13.46%) | 27 (13.24%) | |
| Radiation therapy (RT) site | | | | | |
| | Primary site | 237 (37.32%) | 174 (40.37%) | 63 (30.88%) | 0.021 |
| | Primary site and regional lymph nodes | 398 (62.68%) | 257 (59.63%) | 141 (69.12%) | |
| RT fraction | | | | | |
| | Conventional | 492 (77.48%) | 311 (72.16%) | 181 (88.73%) | <0.001 |
| | Hypofractionated | 143 (22.52%) | 120 (27.84%) | 23 (11.27%) | |
| Localized boost treatments | | | | | |
| | No | 387 (60.94%) | 258 (59.86%) | 129 (63.24%) | 0.416 |
| | Yes | 248 (39.06%) | 173 (40.14%) | 75 (36.76%) | |
| Bolus application | | | | | |
| | No | 275 (43.31%) | 202 (46.87%) | 73 (35.78%) | 0.008 |
| | Yes | 360 (56.69%) | 229 (53.13%) | 131 (64.22%) | |

used, conventional fractionated radiation therapy (CFRT) was administered to 492 patients (77.48%), whereas hypofractionated radiation therapy (HFRT) was utilized for 143 patients (22.52%). Furthermore, a sequential boost was administered to 248 patients (39.06%), and bolus application was used in 360 patients (56.69%).

### Predictive factors associated with ARD severity

The predictive factors of ARD were assessed through univariate and multivariable analyses. In the univariate analysis, significant predictors included age, BMI, diabetes, hypertension, type of surgery (mastectomy *vs.* breast-conserving surgery), adjuvant chemotherapy, hormonal therapy, targeted therapy, RT site (primary site *vs.* primary site and regional lymph nodes), RT fractionation (conventional *vs.* hypofractionated), boost treatment, and bolus application, as detailed in Table 3.

The multivariable analyses identified three independent factors associated with an increased risk of severe ARD, as shown in Table 4. Specifically, patients with a BMI $\geq$ 30 kg/m$^2$ were significantly more likely to develop ARD compared to those with a BMI < 25 kg/m$^2$ (AOR = 2.33; 95% CI [1.36–3.98]; $p$-value = 0.002). Additionally, patients who received a radiation boost had a higher likelihood of developing ARD than those without a boost (AOR = 2.09; 95% CI [1.08–4.06]; $p$-value = 0.029). Similarly, while patients treated with bolus showed an increased risk (AOR = 2.08; 95% CI [1.02–4.24]; $p$-value = 0.044), patients treated without bolus did not present a significant risk. Conversely, two independent factors were associated with a decreased risk of severe ARD: patients who underwent HFRT demonstrated a lower risk of ARD compared to those who received CFRT (AOR = 0.31; 95% CI [0.16–0.57]; $p$ < 0.001). Furthermore, patients who received hormonal therapy showed a lower risk of ARD than those who did not receive hormonal treatment (AOR = 0.60; 95% CI [0.42–0.86]; $p$-value = 0.005). However, neither the RT site (primary site *vs.* primary site and regional lymph nodes; AOR = 0.81; 95% CI [0.41–1.59]; $p$-value = 0.538) nor the application of targeted therapy (targeted therapy *vs.* no targeted therapy; AOR = 0.72; 95% CI [0.43–1.20]; $p$-value = 0.210) led to a significant difference in the ARD rate (Table 4).

## DISCUSSION

ARD is a prevalent and distressing side effect experienced by breast cancer patients undergoing RT. The results of this retrospective cohort study revealed that there were several predictive factors associated with the development of ARD, which can significantly impact treatment outcomes. The nature of the identified risk factors, which included a higher BMI, the use of radiation boosts, and bolus application, suggests that certain patient and treatment characteristics predispose individuals to more severe ARD. Conversely, the finding that HFRT and hormonal therapy mediated protective effects indicates that these treatment modalities can reduce the severity of ARD.

Numerous studies have established a significant association between BMI and the development of ARD in breast cancer patients undergoing RT (*Xie et al., 2021*). Our results are consistent with these findings, demonstrating a significant association between a higher BMI and the incidence and severity of ARD among patients undergoing RT. Specifically,

**Table 3** Predictors of acute radiation dermatitis severity in breast cancer patients (univariate analysis).

| Risk factor | | Maximum grade ≥ 2 | | |
|---|---|---|---|---|
| | | Crude odds ratio | 95% confidence interval | p-value |
| Age (years) | | | | |
| | ≤60 | Ref. | | |
| | >60 | 0.78 | [0.53, 1.13] | 0.188 |
| BMI (kg/m$^2$) | | | | |
| | <25 | Ref. | | |
| | 25–29.90 | 1.26 | [0.87, 1.84] | 0.223 |
| | ≥30 | 2.39 | [1.43, 3.99] | 0.001 |
| Diabetes | | | | |
| | No | Ref. | | |
| | Yes | 1.16 | [0.70, 1.92] | 0.556 |
| Hypertension | | | | |
| | No | Ref. | | |
| | Yes | 1.03 | [0.70, 1.52] | 0.866 |
| Surgery type | | | | |
| | Mastectomy | Ref. | | |
| | Breast-conserving surgery | 0.65 | [0.46, 0.91] | 0.013 |
| Adjuvant chemotherapy | | | | |
| | No | Ref. | | |
| | Yes | 1.37 | [0.92, 2.05] | 0.124 |
| Hormonal therapy | | | | |
| | No | Ref. | | |
| | Yes | 0.59 | [0.42, 0.82] | 0.002 |
| Targeted therapy | | | | |
| | No | Ref. | | |
| | Yes | 0.98 | [0.60, 1.60] | 0.939 |
| Radiation therapy (RT) site | | | | |
| | Primary site | Ref. | | |
| | Primary site and regional lymph nodes | 1.51 | [1.06, 2.16] | 0.021 |
| RT fraction | | | | |
| | Conventional | Ref. | | |
| | Hypofractionated | 0.33 | [0.20, 0.53] | <0.001 |
| Localized boost treatments | | | | |
| | No | Ref. | | |
| | Yes | 0.87 | [0.61, 1.22] | 0.416 |
| Bolus application | | | | |
| | No | Ref. | | |
| | Yes | 1.58 | [1.12, 2.23] | 0.009 |

**Notes.**
The variables were defined based on a review of the literature related to factors that affect acute radiation dermatitis.

**Table 4  Predictors of acute radiation dermatitis severity in breast cancer patients (multiple analyses).**

| Risk factor | | Maximum grade ≥ 2 | | |
|---|---|---|---|---|
| | | Adjusted odds ratio | 95% confidence interval | *p*-value |
| BMI (kg/m$^2$) | | | | |
| | <25 | Ref. | | |
| | 25–29.90 | 1.28 | [0.86, 1.89] | 0.216 |
| | ≥30 | 2.33 | [1.36, 3.98] | 0.002 |
| Hormonal therapy | | | | |
| | No | Ref. | | |
| | Yes | 0.60 | [0.42, 0.86] | 0.005 |
| Targeted therapy | | | | |
| | No | Ref. | | |
| | Yes | 0.72 | [0.43, 1.20] | 0.210 |
| Radiation therapy (RT) site | | | | |
| | Primary site | Ref. | | |
| | Primary site and regional lymph nodes | 0.81 | [0.41, 1.59] | 0.538 |
| RT fraction | | | | |
| | Conventional | Ref. | | |
| | Hypofractionated | 0.31 | [0.16, 0.57] | <0.001 |
| Localized boost treatments | | | | |
| | No | Ref. | | |
| | Yes | 2.09 | [1.08, 4.06] | 0.029 |
| Bolus application | | | | |
| | No | Ref. | | |
| | Yes | 2.08 | [1.02, 4.24] | 0.044 |

**Notes.**
The surgery type was excluded from the multiple analyses to allow focus on the relationship between bolus use and the grading of acute radiation dermatitis.

we found that a BMI ≥ 30 kg/m$^2$ was associated with an increased risk of developing ARD classified as grade ≥ 2 (AOR = 2.33; 95% CI [1.36–3.98]; *p*-value = 0.002). This aligns with other studies that reported that 80% of patients who presented with wet desquamation (grade 2) had a BMI ≥ 25 kg/m$^2$ (*Córdoba, Lacunza & Güerci, 2021*). In the same study, it was observed that 69% of patients who developed grade ≥ 2 acute radiation dermatitis were classified as overweight or obese (BMI > 25 kg/m$^2$), with obese patients having 2.87 times more risk of manifesting radiodermatitis than those with a normal BMI (AOR = 2.87, *p*-value = 0.026). Additionally, in a study conducted among 598 patients, those with a BMI ≥ 24 kg/m$^2$ were found to be more likely to develop dermatitis than those with a BMI < 24 kg/m$^2$ (*Liu et al., 2022*).

Moreover, *Cavalcante et al. (2024)* found that for each unit increase in BMI, the likelihood of developing ARD classified as grade ≥2 increases by 1.14 times (AOR = 1.14; 95% CI [1.04–1.26]; *p*-value = 0.004) (*Cavalcante et al., 2024*). This trend has also been supported by a larger cohort study involving 3,518 patients, which found BMI to be a statistically significant risk factor for grade 2–3 acute dermatitis (AOR = 2.30; 95%

CI [1.28–4.26]; $p < 0.01$) (*Issoufaly et al., 2022*). In terms of treatment regimens, both CFRT and HFRT have shown significant associations with ARD and moist desquamation in relation to BMI ($p$-value = 0.003 and $p < 0.001$, respectively) (*De Langhe et al., 2014*). Additionally, factors such as breast volume and comorbidities may further exacerbate the risk of ARD in overweight patients (*Xie et al., 2021*). In breast cancer patients, particularly those with higher BMI values, the presence of more voluminous skin folds can create a bolus effect during RT, especially in inframammary and axillary regions. This phenomenon occurs because the additional tissue thickness can inadvertently increase the radiation dose delivered to the skin surface, potentially exacerbating the risk of ARD (*Liu et al., 2022*).

Our results indicate that the incidence of ARD was significantly higher in patients receiving boost irradiation (OR = 2.09; 95% CI [1.08–4.06]; $p$-value = 0.029). This finding is consistent with those of previous studies that demonstrated that administering boosts to the tumor bed exacerbated skin reactions in breast cancer patients. For instance, a study involving 502 patients treated with IMRT and a simultaneous integrated boost reported that grade $\geq 2$ ARD occurred significantly more often in patients who underwent IMRT compared to those who received 3DCRT with a sequential boost (29.1% *vs.* 20.1%; $p$-value = 0.02) (*Krug et al., 2020*). However, without boost irradiation, 3DCRT was found to cause more radiation dermatitis than IMRT because The IMRT technique delivers radiation from multiple directions compared to 3DCRT, resulting in greater skin sparing and, consequently, less dermatitis (*Chen et al., 2020*). In another study of 75 patients, it was found that boost treatment was significantly associated with an increased risk of ARD (OR = 4.61; 95% CI [1.33–15.93]; $p$-value = 0.01) (*Abdeltawab et al., 2021*). Additionally, a cohort study involving 489 breast cancer patients revealed that grade $\geq 2$ ARD was significantly more prevalent in those who received a radiation boost compared to those who did not (30.4% *vs.* 13.5%; $p < 0.0001$) (*Eggert, Yu & Rades, 2023*). These findings collectively support the notion that boost treatment is a critical predictor of ARD. However, it is noteworthy that a study involving 1,093 patients found no significant difference in skin outcomes between patients who received simultaneous and sequential boosts; nonetheless, the use of a boost was predictive of edema ($p$-value = 0.02) (*Behroozian et al., 2021*). This discrepance more likely highlights the complexity of radiation-induced skin toxicity and suggest that while boosts generally increase the risk of ARD, individual treatment regimens and patient characteristics also contribute to the varied outcomes (*Schmeel et al., 2020*).

The use of bolus material in post-mastectomy radiation therapy (PMRT) has been a topic of considerable debate due to its dual role in enhancing radiation dose delivery to superficial tissues while simultaneously increasing the risk of ARD (*Wong et al., 2020*). Our findings align with previous research indicating that the application of a bolus significantly increases the incidence of severe ARD (*Parekh et al., 2018*). In our study, patients who received RT with bolus had an AOR of 2.08 (95% CI [1.02–4.24]; $p$-value = 0.044) for developing ARD, reinforcing the notion that bolus use is a critical predictor of this complication. *Krug et al. (2020)* reported that approximately 80% of breast cancer patients developed grade 2–3 ARD following PMRT, and they identified bolus use as a key contributing factor to this outcome. This finding is supported by our analysis, which indicates that bolus application can exacerbate skin toxicity, leading to higher rates of severe ARD. In another

study, PMRT without bolus use resulted in a significantly lower incidence of severe ARD, with only 10% of patients developing grade 2 ARD, and this was achieved without any increase in local recurrence rates (*Eggert, Yu & Rades, 2023*). Additionally, a cohort study involving 1,093 patients found that bolus use was the only factor significantly associated with bleeding complications (*p*-value = 0.02), suggesting that while bolus use may enhance treatment efficacy in terms of local control, it also poses risks for acute toxicities that could complicate patient management (*Behroozian et al., 2021*). Moreover, the advantages of using personalized silicone rubber boluses have been highlighted in other studies; for example, *Chen et al. (2024)* reported that only 16.6% of patients developed grade 2 ARD when utilizing such materials. This suggests that not all bolus applications are equal and that there is a need for personalized approaches based on patient characteristics and treatment goals (*Verma, 2023*).

Our analysis supports the hypothesis that using HFRT does not increase the risk of ARD (*Sinzabakira et al., 2023*), as evidenced by an OR of 0.31 (95% CI [0.16–0.57]; $p < 0.001$) because HFRT delivers a higher radiation dose per fraction, while the total dose of HFRT is lower than that of CFRT, which is associated with reduced occurrences of breast shrinkage, telangiectasia, and breast edema (*Butler-Xu et al., 2019*). This finding aligns with those of previous studies that indicated that patients who underwent HFRT experienced significantly lower rates of acute skin toxicity compared to those who received CFRT. For instance, in a cohort of 339 patients, grade 2 and 3 ARD were reported to occur in 42% and 13% of CFRT patients, respectively, compared to 30% and 7.5% of HFRT patients (*Tortorelli et al., 2013*). The overall acute toxicity rates were notably higher among patients treated with CFRT (81.6%) compared to those treated with moderate HFRT (62.6%), and this difference was statistically significant ($p < 0.001$). Further supporting these findings, a study involving 377 patients demonstrated that those treated with HFRT developed less ARD than those treated with a normofractionated regimen ($p < 0.001$) (*De Langhe et al., 2014*). Additionally, a larger analysis of 1,727 patients found that the incidence of grade $\geq 2$ ARD was significantly lower in a group treated with HFRT compared to a group treated with conventional treatment (OR = 0.34; 95% CI [0.29–0.41]) (*Bruand et al., 2022*). Similarly, another study reported that only 8.9% of patients treated with HFRT experienced grade 2–3 ARD compared to 35.1% of patients treated with CFRT ($\chi^2 = 373.7$; $p < 0.001$) (*Issoufaly et al., 2022*). Despite these consistent findings suggesting that HFRT regimens result in a reduced incidence of ARD, there are instances in which no significant difference was observed in the incidence of ARD between CFRT and HFRT. For example, one study indicated that while CFRT increased the skin dose and was a significant predictor of worse skin reactions, the use of CFRT and HFRT did not yield significant differences in terms of ARD (*Sawanee Nirunsiriphol, 2021*). This discrepancy may stem from variations in the patient populations, treatment protocols, and dosimetric factors across studies. This highlights the need for further research to clarify the conditions under which HFRT may or may not confer protective benefits against ARD (*Parekh et al., 2018*).

In this study, the use of hormone therapy, regardless of type, was associated with a significant decrease in ARD (OR = 0.60; 95% CI [0.42–0.86]; $p$-value = 0.005). This finding contrasts with those of previous studies that investigated the effects of hormone

therapy on acute skin reactions during RT. For instance, a study involving 377 patients found that concomitant hormone therapy was associated with an increased risk of ARD ($p$-value = 0.041), with no significant difference in incidence between the use of aromatase inhibitors and tamoxifen (*De Langhe et al., 2014*). In another study of 104 women, among those who underwent concomitant hormone therapy alongside HFRT, 70.5% developed some form of skin reaction, primarily erythema (*Vieira et al., 2022*). These results suggest that there is a complex relationship between hormone therapy and skin toxicity during RT, which may vary based on treatment protocols and patient characteristics. Additionally, a case report noted the occurrence of radiation recall dermatitis (RRD) induced by tamoxifen, although such cases are rare (*Rhee et al., 2014*). Furthermore, in two retrospective studies, no significant differences were observed in the incidence of radiation toxicity related to the concurrent or sequential use of tamoxifen in breast cancer patients (*Liu et al., 2022*). These contradictory findings highlight the need for further investigation into the role that hormone therapy plays in ARD. While our data suggest a protective effect, other studies indicate potential risks associated with specific hormonal treatments. Future research should aim to clarify these relationships and explore the underlying mechanisms that may contribute to varying outcomes based on hormone therapy type and treatment context (*De Langhe et al., 2014*).

Previous studies have shown that patients who receive targeted therapies, particularly trastuzumab treatment, have a decreased risk of ARD. More specifically, in a study involving 377 patients, (*De Langhe et al., 2014*) demonstrated that those who received trastuzumab had a significantly reduced risk of ARD ($p < 0.001$). Similarly, in their analysis of 75 patients, (*Abdeltawab et al., 2021*) revealed that the use of trastuzumab was associated with a decreased risk of ARD ($p$-value = 0.01). These results suggest that targeted therapies may play a protective role against ARD. However, contrasting evidence exists in the literature. A study involving 598 patients indicated a tendency for ARD to be associated with adjuvant targeted therapy ($p$-value = 0.052) (*Liu et al., 2022*). These findings suggest that while targeted therapies (*e.g.*, use of trastuzumab) may reduce the risk of ARD in some cases, there may be specific contexts or patient populations in which these therapies could potentially contribute to increased skin toxicity (*Anupama, Anuradha & Maka, 2018*). In line with our study's results, which showed that targeted therapies were not significantly associated with the development of grade $\geq 2$ ARD (OR = 0.72, 95% CI [0.43–1.20]; $p$-value = 0.210), it appears that the protective effect of targeted therapies can vary based on treatment regimens and individual patient factors. This variability highlights the complexity of interactions between cancer treatments and their dermatological effects.

Regarding the effect on ARD of directing RT to regional lymph nodes, our results indicate that there was no significant difference in the severity of ARD between the patients who received RT for primary tumors alone and those who received RT for both primary tumors and regional lymph nodes (OR = 0.81; 95% CI [0.41–1.59]; $p$-value = 0.538). This finding aligns with those of previous studies that reported no significant differences in ARD incidence between node-negative and node-positive patients, regardless of the fractionation schedule ($\chi^2 = 0.24$; $p$-value = 0.62) (*Issoufaly et al., 2022*). However, contrasting evidence exists regarding the relationship between nodal involvement and ARD severity. For instance,

 

a study involving 220 patients who underwent breast-conserving surgery found that the severity of ARD was associated with lymph node stage and that patients with positive regional lymph node metastasis were more susceptible to ARD (*p*-value = 0.009) (*Liu et al., 2022*). This suggests that while we did not observe a significant difference in ARD severity based on nodal involvement, there may be specific contexts in which nodal status influences the occurrence of ARD; for example, it may affect the treatment techniques used, an individual patient's characteristics, and the overall radiation dose delivered (*Bennardo et al., 2021*). The results of this study indicate that some factors align with previous research and can be applied in clinical practice by closely monitoring patients with risk factors. However, certain factors show inconsistent results compared to other studies, possibly due to insufficient data and single center. Despite this, the findings remain interesting and beneficial to patients. Therefore, further studies with a larger sample size across multiple centres are needed. Focusing on the differences in the type and dosage of hormonal drugs received by patients may provide valuable insights. These findings could potentially support the role of hormones in reducing acute radiation dermatitis.

## Limitations

The retrospective nature of this study inherently limits our ability to establish causation. Data were collected from existing medical records that were not originally designed for research purposes, which may have led to incomplete or inaccurate information. Moreover, missing data and a reliance on historical records can result in gaps in data collection, especially for variables that are not routinely documented. This can compromise the reliability of findings and introduce bias, as missing information may skew results and interpretations. These limitations emphasize the need for caution when interpreting the results and suggest that further prospective studies are necessary to validate the findings and explore causal relationships more thoroughly. There may be additional risk factors that were not considered in this study, such as genetic markers and exposing the skin to radiation. Furthermore, the limited number of patients with certain characteristics, such as those older than 60 years and those classified as obese, may have introduced bias during the assessment of the association of characteristics with ARD. Finally, this study was conducted at a single center in Thailand. Hence, further studies involving multiple centers across Thailand and other countries are necessary to establish the broader clinical relevance of our findings.

## CONCLUSIONS

The study highlights critical predictive factors for ARD in breast cancer patients undergoing RT, emphasizing the importance of BMI, the treatment modalities applied (*e.g.*, boost and bolus applications), and the effects of fractionation and hormonal therapy on skin outcomes. These findings have implications for clinical practice. They can be used to identify high-risk patients and tailor preventive strategies to mitigate ARD.

## ACKNOWLEDGEMENTS

The authors would like to express their gratitude to the Radiation Oncology Department of Chulabhorn Hospital for their valuable support in providing the data required for this research. Additionally, the authors wish to acknowledge Prof. Dr. Chukiat Viwatwongkasem for his advice and Kristen Sadler, Ph.D., from Scribendi for editing a draft of this manuscript. We used OpenAI's ChatGPT 4.0 for language editing during the preparation of this manuscript.

### Funding

This research project was supported by Chulabhorn Royal Academy. The funders had no role in study design, data collection and analysis, decision to publish, or preparation of the manuscript.

### Grant Disclosures

The following grant information was disclosed by the authors:
Chulabhorn Royal Academy.

### Competing Interests

The authors declare there are no competing interests.

### Author Contributions

- Rattanaporn Nanthong conceived and designed the experiments, performed the experiments, analyzed the data, prepared figures and/or tables, authored or reviewed drafts of the article, and approved the final draft.
- Sunanta Tungfung conceived and designed the experiments, authored or reviewed drafts of the article, and approved the final draft.
- Kamonwan Soonklang conceived and designed the experiments, analyzed the data, authored or reviewed drafts of the article, and approved the final draft.
- Wiriya Mahikul conceived and designed the experiments, performed the experiments, analyzed the data, prepared figures and/or tables, authored or reviewed drafts of the article, and approved the final draft.

### Human Ethics

The following information was supplied relating to ethical approvals (i.e., approving body and any reference numbers):

The study was conducted in accordance with the guidelines of the Declaration of Helsinki. Before the commencement of this study, ethical approval was granted by the Human Research Ethics Committee of Chulabhorn Royal Academy, Thailand (code: EC 055/2567). The Institutional Review Board of all participating institutions approved the study. Data were encrypted prior to the analysis at the statistical office, where each patient was assigned a unique identifier. This identifier eliminates the possibility to trace the

patient's identity. The final protocol of this study, including the final version of the subject informed consent form, had been approved by the Human Research Ethics Committee of Chulabhorn Royal Academy.

## Data Availability

The raw measurements are available in the Supplementary File.

## Supplemental Information

Supplemental information for this article can be found online at http://dx.doi.org/10.7717/peerj.19202#supplemental-information.

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
