# Peer review of "Predictive factors associated with acute radiation dermatitis in patients with breast cancer: a retrospective cohort study"

_PeerJ, doi:10.7717/peerj.19202_

## Round 0.1 · original submission · Major Revisions

Please respond to all the comments of both reviewers, in detail

Reviewer 1 ·

Basic reporting

The authors presented a study with an easy-to-understand design and with sufficient references to support its purpose and need in the field. However, the text should be improved by correcting the English language.

Experimental design

No comment

Validity of the findings

No comment

Additional comments

Below are presented some issues to be amended:
1.Paragraph from Line 99-104 (“Approximately 200-300 breast cancer patients…”) should be inserted in the introduction.
2.To avoid confusion, the authors should insert in the text, where necessary, that chemotherapy was adjuvant (abstract: "...treatment-related variables surgical history, chemotherapy, hormone therapy, targeted therapy...")
3.Concurrent hormonal therapy and adjuvant chemotherapy were in the inclusion criteria? If yes, please mention in the paragraph (line 105-111: "The inclusion criteria used to select patients for this study...").
4.Line 119: Please insert a reference to the statement “Each patients treatment plan was developed using established guidelines.”.
5.Line 121: Please change from “formulated the dose distribution” to “planned the dose distribution”.
6.Line 128: Please rephrase “with or without the addition of a dose boost and/or bolus” to “with or without sequential boost irradiation and/or bolus”. I assume it is a sequential boost irradiation from the fractionation regimen (2 Gy/day). Were there patients also treated with integrated boost? Please also mention it in the text.
7.Table 1: Please insert a table header.
8.Table 2: Please update the table header with the name of columns 1 and 2. Also, please use the term “p-value” for the statistical test coefficient.
9.Line 187: Please mention if there was sequential boost irradiation or integrated boost irradiation (“…a radiation boost was administered…”).
10.Please move Table 2 from Predictive factors associated with ARD severity section to Patient demographics and treatment characteristics section.
11.Table 3 and 4: Please use the term “p-value” for the statistical test coefficient.
12.Line 202-203: Please rephrase “Similarly, compared to those treated without bolus use, those treated with bolus use also showed an increased risk” to “Similarly, while patients treated with bolus showed an increased risk (AOR = 2.08; 95% CI [1.02, 4.24]; p-value = 0.044), patients treated without bolus did not present a significant risk.”.
13.Please use the term “p-value” for the statistical test coefficient throughout the text (line 199: p = 0.002; line 201: p = 0.029; and so on).
14.Line 242: Please insert the measurement unit to BMI.
15.Line 247-249: Please rephrase the sentence.
16.Line 257: Please rephrase “…especially in regions such as the inframammary and axillary areas.” to “especially in inframammary and axillary regions”.
17.Line 261: Please rephrase “…higher in the patients who received boost treatment…” to “…higher in patients receiving boost irradiation…”.
18.Line 264: Please provide additional information regarding 3DCRT vs IMRT for ARD occurrence. Also, the authors must indicate why one technique has a high risk of ARD compared to the other.
19.Line 282-284: I do agree that bolus increase the occurrence of ARD, but based on your presented correlation, the p-value is 0.044 which you can call it moderate significant, not quite “…significantly heightens the incidence of severe ARD…”.
20.Paragraph from line 302-325: It’s a well presented paragraph, but must be discussed why HFRT presents lower risks of ARD compared to CFRT.

·

Basic reporting

Regarding the language of the manuscript, it appears to be grammatically correct and scientifically appropriate for the most part. The following minor corrections are suggested:
a) Line 50: “...poses a significant global...”.
b) Line 54: “... impact on both national and international levels”.
c) Lines 56-57: Instead of the term “milk duct”, use a histological term such as “terminal duct lobular unit (TLDU)”.
d) Line 57: The introduction of the sentence could be rephrased in a more comprehensible manner. For example “A common occurrence of the natural progression of the disease is the spread of carcinomatous cells...”.
e) Line 75: “...influence possible dermatic reactions to RT”.
f) Line 76: Is is unclear what the authors are referring to by using the term “loss of control”.
g) Line 83: “... improve patient care”.
h) Lines 101-104: This particular sentence could be rephrased and/or simplified for it to become more understandable.
i) Lines 179-180: “...most of the patients in the cohort...”.
j) Line 242 (also later in text): There is alternating use of the terms ARD and radiodermatitis. Is it intentional? Are the two terms equivalent or did previous studies expand their field of research to chronic radiation dermatitis as well?
k) Line 244: “...conducted among 598 patients...”.
l) Line 247: “... that for each unit increase...” is written twice.
m) Line 257: “... more voluminous skin folds...”.
n) Lines 261, 278, 284 and 308: Remove “the” before “patients”.
o) Line 276: “This discrepance more likely highlights the complexity...”.
p) Line 283: “... significantly increases the incidence...”.

Considering the literature reviewed throughout the manuscript, the authors provide extensive references to previous studies with designs similar to the current study, while also addressing discrepancies between their results. This is perhaps the strongest feature of the manuscript; however, some additions would enhance the context of the study:
a) Lines 56-7: The discussion on the histogenesis of breast carcinoma is overly vague. A brief paragraph outlining key molecular events in the natural history of the most common types of breast carcinoma would provide valuable context.
b) Similarly, the discussion on the pathophysiology of Acute Radiation Dermatitis (ARD) is almost nonexistent. Lee et al. (Ann Dermatol. 2009 Nov 30;21(4):358–363. doi: 10.5021/ad.2009.21.4.358) present excellent literature on the pathophysiological mechanisms of ARD in their introduction. While it is understandable that parameters with potential prognostic or predictive value are prioritized based on availability, it is important to acknowledge or hypothesize the underlying pathophysiological basis of any relevant findings.
c) Line 72: Do different radiation techniques (e.g., CyberKnife, TomoTherapy, etc.) result in varying outcomes regarding the presentation of ARD?
d) Line 73: “... are also contributing factors”. No source privded for that claim.
e) Lines 85-86: “This variability... factors that have not yet been identified.” In the discussion section, the referenced studies indicate that these factors have been identified, although there is no consensus on which of them have prognostic or predictive significance. For accuracy, the authors should rephrase this statement and emphasize that the value of their study lies in investigating a field with contradictory results.
f) Lines 106-107: “... completed the entire course of RT...”. The authors should mention the range of the various radiation therapy (RT) courses.
g) Line 119: “... established guidelines.” Are the authors referring to national or international guidelines? It would be helpful to elaborate on how the RT plan is designed and whether planning parameters not examined in this study could introduce confounding factors.
h) Lines 135-136: “... is the most widely used tool for this purpose.” The authors support this claim with a reference from 1995. They should consider citing more recent studies on acute radiation dermatitis (ARD) to prove that point.
i) Given the conflicting evidence from previous studies and the findings of this study, the authors are encouraged to discuss how they would use existing knowledge to personalize radiation therapy treatment planning for breast cancer patients.

A slight deviation in article structure (i.e. a separate "Limitations" subsection between "Discussion" and "Conclusion" sections) is justified. No further comment on tables.

Raw data of the study are attached. No further comment required.

Results are relevant to the hypothesis of the study. No further comment required.

Experimental design

The research in question aligns with the Aims and Scope of PeerJ.

Regarding the definition, relevance and meaningfulness of the research question, the technical standards as well as the robust description and replicability of methodology, the present study seems to be in line with previous, similarly designed studies, thus exhibiting adequacy in all the above fields.

Considering sufficient statement on how the current research fills an exisitng knowledge gap, kindly see the previous section of this review.

Finally, all ethical approval statements are attached. No further comment on ethical standards.

Validity of the findings

The research in question appears to be the latest in a series of publications investigating predictive factors related to the presentation of ARD in breast cancer patients undergoing RT. Considering the discrepancies in the results of previous studies, the present study is justified as it contributes its own conclusions to the existing body of scientific evidence.

All data used to draw the conclusions of the present study are attached and their statistical analysis follows the methodology of previous studies. No further comment required.

Conclusions are well stated, yet the article would benefit if the authors elaborated on how they would utilize their conclusions in everyday clinical practice (see also the first sextion of this preview).

Additional comments

The present study's design appears to be overally straightforward, whereas its findings are in accordance with the findings of the majority of previous similar studies. No real question could arise regarding its methodological approach or the validity of its results.
Alongside with language correction and clarification, the improvements suggested in the present review serve to provide more context, mostly on pathophysiology and the wider aspect of clinical practice. Once in place, the manuscript in question could be accepted for publication.

---

## Round 0.2 · accepted · Accept

Dear Authors
Thank you for addressing all the concerns of the referees.
Congratulations

Reviewer 1 ·

Basic reporting

I would like to thank the authors for considering and addressing all submitted comments. The article has been edited and corrected to suit the English language. The tables have also been corrected.
All comments have been amended accordingly, therefore I propose accepting the article in its current format.

Experimental design

No comment

Validity of the findings

No comment